# Neuromuscular screening and cognitive function in older adults: A cross-sectional exploratory study

**Petr Schlegel[1], Zdeněk Zadák[2], Radka Dostálová[1], Adrián Agricola[1]***

**1** Department of Physical Education and Sport, Faculty of Education, University of Hradec Králové, Czech Republic, **2** University Hospital Hradec Králové, Hradec Králové, Czech Republic

* adrian.agricola@gmail.com

## Abstract

Age-related decline in both cognitive and neuromuscular function negatively affects independence and quality of life in older adults. Understanding the interplay between cognitive performance and physiological markers may support early identification of functional deterioration. This cross-sectional study examined associations between cognitive function and selected physiological parameters in community-dwelling adults aged 65 years and older. Cognitive performance was assessed using the Montreal Cognitive Assessment (MoCA). Additional measures included handgrip strength, body composition (bioelectrical impedance analysis), physical activity level (IPAQ-SF), and electrically evoked neuromuscular responsiveness using a custom-built diagnostic device. The average MoCA score was $25.5 \pm 2.1$, with 50% of participants scoring below the commonly used clinical threshold. Handgrip strength was strongly correlated with skeletal muscle mass ($r = 0.64$), but only weakly associated with cognitive function ($r = 0.22$). Neuromuscular responsiveness, specifically the slope at 60 Hz stimulation, showed a moderate negative correlation with MoCA scores ($r = -0.35$), accounting for 20% of its variance ($R^2 = 0.20$; $p = 0.088$). Physical activity levels were generally low and not significantly related to cognitive or muscular parameters. Electrically evoked neuromuscular markers may serve as complementary indicators to traditional strength assessments and could support early detection of neuromuscular and cognitive decline in older adults.

## Introduction

The global population is undergoing a profound demographic transformation, with projections indicating that by 2050, individuals aged 65 years and older will comprise nearly 30% of the European population. This demographic shift is paralleled by an alarming behavioral trend: in 2022, approximately 31% of adults worldwide—equivalent to 1.8 billion people—failed to meet the recommended levels of physical activity.

**Data availability statement:** All relevant data are within the paper and its Supporting Information files.

**Funding:** The author(s) received no specific funding for this work.

**Competing interests:** The authors have declared that no competing interests exist.

In Europe, the prevalence of physical inactivity among older adults has been reported to range from 5% to 29% [1]. Validated self-report measures such as the International Physical Activity Questionnaire (IPAQ) have been widely used to assess physical activity and sedentary behavior across populations [2,3]. However, their limitations highlight the need for more robust, multidimensional screening approaches [4].

Physical inactivity contributes significantly to the rising prevalence of both functional limitations and cognitive impairments among older adults. Cognitive aging is characterized by gradual declines in executive function, memory, and processing speed, potentially progressing to mild cognitive impairment or dementia [5–7]. In this context, the Montreal Cognitive Assessment (MoCA) has gained recognition as a sensitive tool for detecting early cognitive changes [5,7].

Neuromuscular function also deteriorates with age, with significant declines observed in muscle mass, strength, and power [8–10]. These changes have functional implications, contributing to mobility limitations, fear of falling, and reduced quality of life. Although muscle mass and strength are often correlated, studies indicate that the decline in strength (dynapenia) is not solely explained by loss of muscle mass (sarcopenia), suggesting additional neuromuscular factors at play [8,11].

Furthermore, anthropometric measures and body composition parameters have shown predictive value for muscle strength, especially handgrip strength [12,13]. Recent work has demonstrated notable differences in body composition between older adults with and without cognitive disorders, indicating potential diagnostic relevance [14,15].

Mounting evidence supports a bidirectional association between cognitive and physical function. Greater muscle strength has been consistently linked with better cognitive outcomes in older populations [16,17]. Several meta-analyses confirm that physical exercise interventions, particularly those combining aerobic or resistance training with cognitive challenges, significantly enhance executive function and global cognition [18–21]. Innovative approaches such as the integration of transcranial direct current stimulation with exercise have also demonstrated promise [22].

Notably, combined physical-cognitive programs delivered in socially enriched environments have been shown to improve both cognitive functioning and quality of life, underscoring the importance of holistic and context-sensitive interventions [23]. These findings reinforce the value of multimodal strategies in maintaining cognitive resilience and delaying age-associated decline.

While grip strength remains a widely used proxy for overall muscular function due to its accessibility, it does not fully capture the complexity of neuromuscular decline. Performance-based tests such as the one-minute sit-to-stand test offer additional insights into lower-limb endurance and coordination [24,25]. Yet, even these may not reflect early neuromuscular dysfunction, highlighting the need for high-resolution screening tools that incorporate both functional and morphological indicators.

Consequently, the development of integrated, non-invasive diagnostic protocols that address both neuromuscular and cognitive domains is increasingly emphasized in gerontological research and practice [26]. Such tools have the potential to detect subtle functional deterioration, inform individualized interventions, and monitor progress across time.

Accordingly, the present preliminary study aimed to investigate associations between cognitive performance and a comprehensive set of physiological parameters, including muscle morphology, neuromuscular responsiveness, muscle strength, anthropometry, and body composition in a sample of community-dwelling older adults. The study also explored the feasibility of incorporating a novel neuromuscular stimulation protocol into a multidimensional assessment framework, with the aim of generating preliminary, hypothesis-driven insights into age-related neuromuscular–cognitive associations.

## Methods

### Study design

This study was designed as an initial, exploratory investigation to assess the feasibility of integrating neuromuscular stimulation protocols into cognitive screening in older adults. The primary objective was to identify potential relationships between specific neuromuscular indicators (e.g., slope of the force–intensity response) and cognitive performance (MoCA score), in order to inform future hypothesis-driven research. While no formal power analysis was conducted due to the exploratory nature of the study, the sample size was based on similar pilot investigations and enabled initial correlational and regression analyses involving up to three predictors [27].

### Participants

The sample consisted of 20 older adults (16 women, 4 men). Participants were aged 66–77 years at the time of assessment. Participants were students enrolled in a lifelong learning program at a public university and were recruited using a convenience sampling approach. A larger group of older adults within the program was informed about the study, and the first individuals who expressed interest and met the inclusion criteria were enrolled. Although the sample was not evenly balanced in terms of sex, all participants completed the full assessment protocol, with no dropouts. Inclusion criteria were age ≥ 65 years, absence of acute illness or injury, and the ability to complete both physical and cognitive assessments. No formal clinical screening for neurological, psychiatric, cardiovascular, or neuromuscular conditions was performed. Participants were recruited from 17/02/2025–15/04/2025. The study protocol was approved by the Ethics Committee of the relevant institution, and all participants provided written informed consent prior to data collection.

### Measures and procedures

**Neuromuscular function assessment.** Objective assessment of muscle contractile properties was conducted using a custom-built device in combination with surface electrical stimulation. The procedure was approved by the Ethics Committee of the relevant medical institution. Bipolar surface electrodes were placed over the motor point of the tibialis anterior muscle within the anterior compartment of the lower leg (pretibial muscle group). Electrical stimulation was delivered using biphasic current pulses with a pulse duration of 200 µs via a constant-current stimulator. The initial stimulation intensity was set to 30 mA and increased in 5 mA increments according to individual tolerance. Each stimulation bout lasted 4 s.

The custom-built neuromuscular assessment device, developed by a certified medical device manufacturer (MEB-STER, Ltd.), was calibrated in a development laboratory and validated in a sample of 192 healthy subjects. Validation analyses demonstrated a mean relative error of 6.39% for the slope ratio of fast- and slow-twitch fiber stimulation, 8.71% for the slope of fast-twitch fiber stimulation values, and 9.67% for the slope of slow-twitch fiber stimulation values.

The resulting torque during ankle dorsiflexion was recorded using an integrated torsion sensor. Torque values were subsequently converted to force (in Newtons) based on the fixed lever arm length of the foot fixation apparatus. The primary outcome was the maximum force generated during the stimulation protocol. For each participant, a force–intensity curve was constructed by plotting maximal force values against stimulation intensities. The average difference in force per 10 mA increment was calculated and normalized to 1 mA steps to estimate the slope of the linear portion of the curve, representing neuromuscular responsiveness.

The following parameters were derived and used for further analyses:

• Slope of the force–intensity curve (N/mA) at 60 Hz stimulation frequency

• Slope of the force–intensity curve (N/mA) at 16 Hz stimulation frequency

• Ratio of the slopes at 60 Hz and 16 Hz (as an index of neuromuscular adaptation profile)

Stimulation frequencies of 16 Hz and 60 Hz were selected to capture different aspects of neuromuscular activation. The lower frequency represented a low-frequency neuromuscular response, whereas the higher frequency elicited a more pronounced high-frequency contractile response, consistent with the exploratory nature of the study.

### Cognitive function assessment

Cognitive performance was assessed using the Montreal Cognitive Assessment (MoCA), a widely validated screening instrument designed to detect mild cognitive impairment in older adults. The MoCA evaluates multiple cognitive domains, including short-term memory, executive function, attention, language, visuospatial abilities, and orientation, with a maximum total score of 30 points. A score below 26 is commonly used as a clinical threshold for potential cognitive impairment. The assessment was administered in a quiet, well-lit room by a trained examiner following standardized instructions. All participants completed the paper-based version of the test in a single session lasting approximately 10–15 minutes.

### Handgrip strength assessment

Handgrip strength was assessed using a calibrated digital dynamometer (Kern MAP 80K1S, Germany) following a standardized protocol. Participants were tested in a standing position with the shoulder adducted and neutrally rotated, elbow flexed at 90°, forearm in a neutral position, and wrist in slight extension. Three maximal-effort trials were performed with the dominant hand, with brief rest intervals between attempts. The highest recorded value (in kilograms) was used for subsequent analyses.

### Body composition assessment

Body composition parameters, including fat mass (FM), skeletal muscle mass (SMM), fat-free mass (FFM), and segmental lean mass, were assessed using bioelectrical impedance analysis (BIA) with a multi-frequency analyzer (BW3A2, InBody Ltd., Amsterdam, Netherlands). The device uses direct segmental measurement with multiple frequencies (5–1000 kHz) and eight-point tactile electrodes to estimate body composition. Prior to testing, participants were instructed to fast for at least 8 hours and to refrain from consuming alcohol, caffeine, or engaging in intense physical activity for 24 hours. Measurements were performed with participants lying supine on a standard examination bed, with electrodes placed on their wrists and ankles, following the standardized procedures recommended by the manufacturer. All assessments were conducted by trained staff using the same device under controlled conditions. The device was calibrated prior to data collection. The resulting values were expressed in absolute terms (kg) and indexed to height squared (e.g., fat mass index, skeletal muscle index) for further analysis.

### Physical activity assessment

Self-reported physical activity levels were assessed using the short form of the International Physical Activity Questionnaire (IPAQ-SF), a validated instrument for estimating habitual activity levels in adults across diverse populations. The questionnaire captures the frequency (days/week) and duration (minutes/day) of physical activity in three intensity categories: walking, moderate, and vigorous, as well as sedentary time (sitting).

Total physical activity was calculated in MET-minutes per week, using standard metabolic equivalent (MET) values assigned to each intensity level (walking = 3.3 METs, moderate = 4.0 METs, vigorous = 8.0 METs). For further analysis, both continuous data (total MET-min/week) and categorical classification (low, moderate, high physical activity) were used based on established IPAQ scoring protocols.

## Data analysis

All statistical analyses were conducted using IBM SPSS Statistics (version 28.0; IBM Corp., Armonk, NY, USA). Given the exploratory design and limited sample size (n = 20), non-parametric and robust analytical methods were applied. Descriptive statistics (mean, standard deviation, range, coefficient of variation, and interquartile range) were calculated for all key physiological and cognitive variables.

Associations between continuous variables were assessed using Spearman's rank correlation coefficients, which are appropriate for non-normally distributed data and small samples. Discrete variables were not included in correlational analyses. To explore potential predictors of global cognitive performance (MoCA total score), two multiple linear regression models were constructed and compared. The first model included neuromuscular responsiveness indicators (slope at 60 Hz and slope ratio), grip strength, and fat mass index (FMI). In the second model, the slope ratio was replaced with the estimated maximal evoked force. All models included an intercept term and were examined for multicollinearity. Observed variability in several parameters reflects expected inter-individual heterogeneity within the older adult sample rather than measurement unreliability. As no a priori criteria for data exclusion were met, no outliers were removed from the analyses.

The threshold for statistical significance was set at $p < 0.05$. Results with p-values between 0.05 and 0.10 were interpreted as statistical trends, in accordance with the exploratory nature of the study.

## Results

### Descriptive statistics

Participants exhibited relatively high levels of body fat mass (BFM: 26.7 ± 11.6 kg) and percent body fat (PBF: 29.8 ± 7.6%) with substantial inter-individual variability. Skeletal muscle mass (SMM: 26.6 ± 5.3 kg) and fat-free mass index (FFMI: 17.3 ± 2.0 kg/m$^2$) fell within expected age-related ranges, while fat mass index (FMI: 9.5 ± 4.2 kg/m$^2$) and waist-to-hip ratio (WHR: 0.97 ± 0.09) demonstrated substantial inter-individual variability.

Mean handgrip strength was 31.6 ± 6.5 kg (range: 23.8–44.7 kg). Cognitive screening using the MoCA revealed a mean total score of 25.5 ± 2.1 (range: 22–30), with 50% of participants scoring below the clinical cut-off of 26 points. Performance was highest in the orientation domain and lowest in visuospatial processing (cube drawing).

Self-reported physical activity levels, assessed via IPAQ-SF, indicated low engagement across the sample (mean total MET-min/week: 125.4 ± 82.2; range: 6.6–334.6), with all participants falling into the "low" activity category per standardized criteria (Table 1).

### Neuromuscular responsiveness

Force–intensity curves were derived at stimulation frequencies of 16 Hz and 60 Hz. At 16 Hz, the mean slope was 28.2 ± 19.6 N/mA (range: 1.0–64.1), while at 60 Hz the slope increased to 44.2 ± 24.8 N/mA (range: 1.0–97.9). The mean slope ratio (60 Hz/16 Hz) was 1.44 ± 0.45, with values ranging from 0.97 to 3.03. These findings suggest high variability in neuromuscular adaptation capacity, consistent with age-related heterogeneity in motor unit recruitment and transmission efficiency.

The coefficient of variation (CV) for the slope at 16 Hz was 0.70, and 0.56 for the slope at 60 Hz, indicating substantial relative dispersion. Interquartile ranges further confirmed wide variability within the central portion of the sample, reinforcing the value of individualized neuromuscular assessment in older adults (Fig 1).

**Table 1. Summary of anthropometric, functional, and cognitive parameters.**

|  | Mean | STD | Min | Max |
|---|---|---|---|---|
| **BMI** | 26.78 | 5.42 | 18.4 | 44.4 |
| **BFM** | 26.69 | 11.58 | 11.9 | 64.4 |
| **SMM** | 26.62 | 5.29 | 19.1 | 38.9 |
| **FFMI** | 17.28 | 1.97 | 14.1 | 20.8 |
| **FMI** | 9.51 | 4.15 | 4.3 | 23.7 |
| **PBF** | 29.76 | 7.55 | 18.4 | 44.4 |
| **WHR** | 0.97 | 0.09 | 0.83 | 1.19 |
| **Grip_strength** | 31.62 | 6.47 | 23.8 | 44.7 |
| **MET_total** | 125.43 | 82.25 | 6.6 | 334.6 |
| **MoCA_total** | 25.45 | 2.09 | 22 | 30 |

**Note.** BMI = Body Mass Index, BFM = Body Fat Mass, SMM = Skeletal Muscle Mass, FFMI = Fat-Free Mass Index, FMI = Fat Mass Index, PBF = Percent Body Fat, WHR = Waist-Hip Ratio, MET = Metabolic Equivalent of Task, MoCA = Montreal Cognitive Assessment

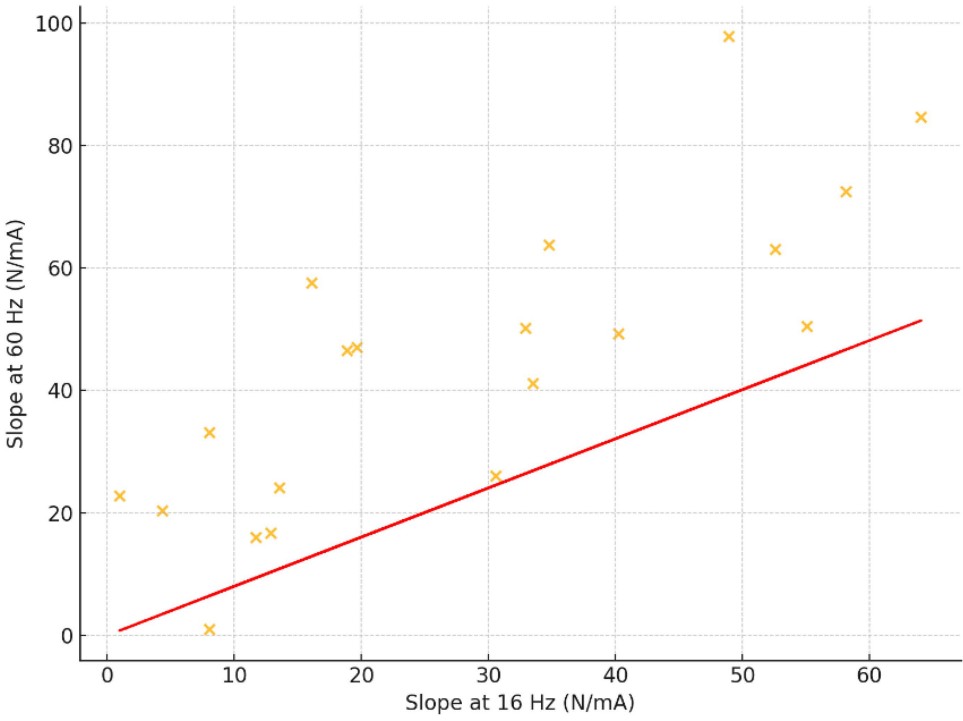

**Fig 1. Scatterplot showing the relationship between neuromuscular slope values at 16 Hz and 60 Hz stimulation frequencies.**

## Correlational and regression analyses

A strong positive correlation was observed between slopes at 16 Hz and 60 Hz (r = 0.85, p < 0.01), indicating consistent neuromuscular responsiveness across stimulation frequencies. Handgrip strength showed a robust association with skeletal muscle mass (r = 0.64) and FFMI (r = 0.45), underscoring the functional relevance of muscle tissue quantity. The neuromuscular slope ratio exhibited a moderate inverse correlation with MoCA scores (r = −0.35), suggesting a potential

link between neuromuscular adaptation profile and cognitive function. Other anthropometric parameters such as FMI and PBF showed weak or inconsistent associations with either cognitive or strength-related variables (Table 2).

Two multiple linear regression models were constructed to predict MoCA performance. In the first model, which included slope60, slope ratio, grip strength, and FMI, the total explained variance was 20.2% ($R^2 = 0.202$). Slope ratio emerged as a marginally significant predictor ($\beta = -2.07$, $p = 0.083$). In the alternative model, where the slope ratio was replaced with estimated maximal evoked force, the explained variance remained similar ($R^2 = 0.197$), with both slope60 ($\beta = 0.50$, $p = 0.082$) and maximal force ($\beta = -0.51$, $p = 0.088$) showing trends toward statistical significance (Table 3).

## Discussion

This exploratory study investigated associations between cognitive performance and various physiological parameters in community-dwelling older adults. The findings highlight marked inter-individual variability in neuromuscular responsiveness and body composition, a strong correlation between skeletal muscle mass and grip strength, and a moderate association between neuromuscular function and cognitive performance. In contrast, self-reported physical activity levels were consistently low and showed no significant relationship with either muscular or cognitive outcomes. These preliminary findings support the conceptual feasibility of incorporating neuromuscular stimulation-based measures into multidimensional research frameworks aimed at understanding neuromuscular–cognitive interactions in aging [8,17].

**Table 2. Spearman correlation matrix of selected body composition, neuromuscular, physical function, and cognitive variables.**

| | MoCA | Total_MET | Grip_strength | Slope16 | Slope60 | Ratio | SMM | FFMI | FMI | PBF | WHR |
|---|---|---|---|---|---|---|---|---|---|---|---|
| **MoCA** | 1 | −0.18 | −0.03 | 0.04 | 0.06 | −0.35 | 0.22 | 0.04 | 0.09 | 0.09 | −0.1 |
| **Total_MET** | −0.18 | 1 | 0.01 | 0.2 | 0.08 | 0.09 | 0.29 | 0.34 | 0.16 | 0.37 | 0.11 |
| **Grip_strength** | −0.03 | 0.01 | 1 | 0.11 | 0.11 | −0.11 | 0.64 | 0.45 | 0.03 | 0.14 | 0.34 |
| **Slope16** | 0.04 | 0.2 | 0.11 | 1 | 0.85 | −0.02 | 0.38 | 0.26 | −0.21 | −0.27 | −0.06 |
| **Slope60** | 0.06 | 0.08 | 0.11 | 0.85 | 1 | 0.27 | 0.32 | 0.21 | −0.27 | −0.31 | −0.19 |
| **Ratio** | −0.35 | 0.09 | −0.11 | −0.02 | 0.27 | 1 | −0.29 | −0.18 | −0.22 | −0.08 | −0.26 |
| **SMM** | 0.22 | 0.29 | 0.64 | 0.38 | 0.32 | −0.29 | 1 | 0.86 | 0.37 | 0.47 | 0.6 |
| **FFMI** | 0.04 | 0.34 | 0.45 | 0.26 | 0.21 | −0.18 | 0.86 | 1 | 0.45 | 0.59 | 0.5 |
| **FMI** | 0.09 | 0.16 | 0.03 | −0.21 | −0.27 | −0.22 | 0.37 | 0.45 | 1 | 0.82 | 0.68 |
| **PBF** | 0.09 | 0.37 | 0.14 | −0.27 | −0.31 | −0.08 | 0.47 | 0.59 | 0.82 | 1 | 0.6 |
| **WHR** | −0.1 | 0.11 | 0.34 | −0.06 | −0.19 | −0.26 | 0.6 | 0.5 | 0.68 | 0.6 | 1 |

**Note.** Values represent Spearman's rank correlation coefficients. MoCA = Montreal Cognitive Assessment; MET = Metabolic Equivalent of Task; SMM = Skeletal Muscle Mass; FFMI = Fat-Free Mass Index; FMI = Fat Mass Index; PBF = Percent Body Fat; WHR = Waist-to-Hip Ratio.

**Table 3. Comparison of multiple linear regression models (Model 1 with slope ratio; Model 2 with maximal force estimate) predicting MoCA score.**

| Predictor | Model_with_Ratio_β | Model_with_Ratio_p | Model_with_MaxForce_β | Model_with_MaxForce_p |
|---|---|---|---|---|
| **Slope60** | 0.013 | 0.56 | 0.504 | 0.082 |
| **Ratio** | −2.067 | 0.083 | | |
| **FMI** | −0.009 | 0.942 | −0.039 | 0.758 |
| **Grip_strength** | −0.042 | 0.607 | −0.051 | 0.535 |
| Max_force_estimate | | | −0.506 | 0.088 |
| | 0.202153 | | 0.197263 | |

**Note.** β = standardized regression coefficient. FMI = Fat Mass Index; MoCA = Montreal Cognitive Assessment; Slope60 = slope of force–intensity curve at 60 Hz; Max force = estimated maximal evoked force. Model 1 includes Slope60 and Ratio; Model 2 includes Slope60 and Max force.

In line with prior studies, we observed a robust correlation between muscle mass and grip strength (r = 0.64), which affirms the role of muscle quantity as a determinant of functional capacity in aging populations [9,12]. However, the only modest relationship observed between grip strength and cognitive performance suggests that isolated strength-based metrics may not sufficiently capture the complexity of neurocognitive decline. This is consistent with research indicating that cognitive–muscle interactions are likely mediated by central mechanisms, including sensorimotor integration, cortical control, and neuromuscular recruitment patterns [11,14].

A noteworthy finding of the present study is the moderate negative correlation between neuromuscular slope ratio (Slope60/Slope16) and MoCA score (r = –0.35), implying that diminished responsiveness to high-frequency electrical stimulation may be associated with poorer cognitive function. Although this relationship did not reach statistical significance, it echoes previous work suggesting that neuromuscular integrity may reflect underlying brain health [17,22]. Interpretation of slope-based parameters, however, warrants caution; a high slope ratio may reflect reduced force generation at lower frequencies rather than enhanced output at higher frequencies. Future studies should thus incorporate raw evoked force values and motor unit analyses to better capture physiological mechanisms.

When the slope ratio was substituted with estimated maximal evoked force, regression analysis revealed a positive trend with MoCA scores (p = 0.088), supporting the hypothesis that greater force-generating capacity may be indicative of more preserved cognitive function. This aligns with growing evidence that the neuromuscular system is tightly linked to cognitive trajectories in aging, particularly in domains related to executive control and motor-cognitive integration [18,20].

In contrast, no significant associations were found between total physical activity, measured via IPAQ (MET-min/week), and cognitive or muscular outcomes. These results are consistent with prior research indicating that self-reported activity measures may lack the sensitivity needed to detect subtle differences in sedentary older adults [2,4]. Objective tools such as accelerometry, gait speed, or dual-task paradigms may be more appropriate for capturing the behavioral correlates of brain health in this population [26].

Of additional interest is the absence of a significant correlation between grip strength and electrically evoked Slope60. This dissociation likely reflects the segmental specificity of neuromuscular function. Previous studies have similarly reported weak associations between upper limb strength and lower limb muscle power, highlighting the limitations of using grip strength as a general marker of global neuromuscular capacity [24,25].

While the present study is limited by its small sample size and exploratory design, the observed trends—particularly the moderate associations between neuromuscular function and cognition—are hypothesis-generating. For instance, the observed effect size (r = –0.35) would require approximately 60 participants to detect a statistically significant effect with adequate power (α = 0.05, power = 0.80), assuming a medium effect (Cohen's $f^2 \approx 0.15$). These data can thus inform the design of future studies seeking to validate neuromuscular–cognitive associations in aging.

Our findings suggest that neuromuscular responsiveness and muscle morphology may provide valuable insights into cognitive aging, beyond what is captured by traditional strength or activity measures. While preliminary, this study supports the feasibility of incorporating neuromuscular stimulation protocols into broader multidimensional screening frameworks aimed at early detection of age-related functional decline. Future research should replicate these findings in larger, sex-balanced, and ethnically diverse populations. Establishing normative values and clinically meaningful thresholds for stimulation-based neuromuscular parameters will be essential to determine their utility in screening, risk stratification, and the evaluation of lifestyle interventions targeting healthy cognitive aging.

The present study is limited by its small sample size and convenience sampling from a university lifelong-learning cohort, which likely represents a relatively healthy and cognitively preserved subgroup of older adults. In addition, the sex imbalance in the sample precluded sex-specific analyses and may have influenced the observed associations. Consequently, the results should not be interpreted as evidence for clinical screening or early detection. Rather, the observed associations should be viewed as hypothesis-generating and intended to inform the design of future, adequately powered studies.

## Conclusions

This exploratory study provides preliminary evidence that neuromuscular responsiveness, as assessed through electrically evoked force parameters, may be a relevant physiological marker in the context of cognitive aging. While handgrip strength was strongly associated with skeletal muscle mass, it showed only weak correlations with cognitive performance. In contrast, neuromuscular indicators—particularly those derived from stimulation-based protocols—demonstrated more consistent associations with global cognitive function. The lack of significant relationships between self-reported physical activity and both cognitive and neuromuscular measures suggests that commonly used questionnaires such as the IPAQ-SF may not adequately reflect functional status in sedentary older adults. These findings underscore the potential value of integrating objective neuromuscular diagnostics into multidimensional assessment frameworks aimed at early identification of age-related decline.

## Supporting information

**S1 File. Data_Body_Composition.**
(XLSX)

**S2 File. Data_MoCa_ IPAQ_Handgrip_strenght.**
(XLSX)

**S3 File. Data_Neuromuscular_function.**
(XLSX)

## Author contributions

**Conceptualization:** Petr Schlegel, Zdeněk Zadák, Radka Dostálová, Adrian Agricola.

**Data curation:** Petr Schlegel, Zdeněk Zadák, Adrian Agricola.

**Formal analysis:** Petr Schlegel, Zdeněk Zadák, Radka Dostálová, Adrian Agricola.

**Investigation:** Petr Schlegel, Radka Dostálová, Adrian Agricola.

**Methodology:** Petr Schlegel, Zdeněk Zadák, Radka Dostálová, Adrian Agricola.

**Project administration:** Petr Schlegel.

**Resources:** Petr Schlegel.

**Supervision:** Petr Schlegel, Zdeněk Zadák.

**Validation:** Petr Schlegel.

**Visualization:** Petr Schlegel, Zdeněk Zadák, Radka Dostálová, Adrian Agricola.

**Writing – original draft:** Petr Schlegel, Adrian Agricola.

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
