## [Decision Letter · Decision Letter 0]

7 Jan 2026

Dear Dr. Agricola,

Thank you for submitting your manuscript to PLOS ONE. After careful consideration, we feel that it has merit but does not fully meet PLOS ONE’s publication criteria as it currently stands. Therefore, we invite you to submit a revised version of the manuscript that addresses the points raised during the review process.

We look forward to receiving your revised manuscript.

Kind regards,

Hiroki Annaka

Academic Editor

PLOS One

Reviewers' comments:

Reviewer's Responses to Questions

**Comments to the Author**

1. Is the manuscript technically sound, and do the data support the conclusions?

Reviewer #1: No

Reviewer #2: Yes

2. Has the statistical analysis been performed appropriately and rigorously?

Reviewer #1: N/A

Reviewer #2: I Don't Know

3. Have the authors made all data underlying the findings in their manuscript fully available?

Reviewer #1: Yes

Reviewer #2: Yes

4. Is the manuscript presented in an intelligible fashion and written in standard English?

Reviewer #1: Yes

Reviewer #2: No

Reviewer #1: Neuromuscular Screening and Cognitive Function in Older Adults is an interesting exploratory cross‑sectional study but it has important limitations in design, reporting, and interpretation that need addressing from authors before it fits journal.

1. It has a very small, convenience sample (n=20), all from a university lifelong-learning course, severely limits external validity and makes claims about “early detection” or “screening” premature and overstated.

2. Sample size justification is a concern, citing “similar pilot investigations” without references or target effect size is weak.

3. Eligibility criteria are minimal with no screening for neurological, psychiatric, major cardiovascular, or neuromuscular conditions that can directly affect both cognition and muscle function

4.Methods : there is no detail on electrode placement landmarks, pulse train structure, exact torque sensor specs, or calibration/verification of the custom device. Is the device checked for Reliability and Validity? Add a substantially more detailed Methods subsection for the neuromuscular device and stimulation protocol, sufficient to reproduce the setup.

5. Results : The statement that fat mass and percent body fat are “elevated compared to normative values” is not backed by explicit normative references or statistical comparisons and therefore sounds speculative.

Reported “mean total MET‑min/week: 125.4 ± 82.2 (range 6.6–334.6)” is orders of magnitude lower than standard IPAQ‑SF MET‑min/week values (which typically run in the thousands), suggesting either that units are not standard or that a transformation has been applied.

Please clarify:

Exactly how IPAQ‑SF data were processed and whether the standard MET‑constants and summation procedures were followed.

The units and scaling used in Table 1 (are these total MET‑hours, averages per day, or rescaled values?).

6. Please: Provide explicit numbers on those invited, declined, and excluded (if available), consistent with STROBE for observational studies.

Reviewer #2: This is a highly intriguing prospective study that establishes an association between two critical factors influencing disordered aging in older people—cognitive impairment and muscle function decline—while incorporating body composition as a control. However, potential issues with sample size and experimental design reliability resulted in largely non-significant findings. The study merits publication.

Suggestions for revision:

1. Describe the basic characteristics of the subjects, including inclusion criteria, age range, and sociodemographic attributes of the population;

2. Justify the selection of 16 Hz and 60 Hz electrical stimulation frequencies.

3. Ensure appropriate methods for correlating continuous and discrete variables;

4. Due to excessively high standard deviation values, data reliability analysis is required, and unreliable data should be excluded.

5. While sample size requirements are less stringent for exploratory studies, statistical analysis remains necessary to ensure the reliability of the results.

6. Gender factors persist.

Research Methodology Questions:

1. What is a “custom-built” device?

2. Why was the tibialis anterior muscle selected as the test site?

3. What is the anatomical location for “targeting the motor point within the anterior compartment of the lower leg”?

**Do you want your identity to be public for this peer review?** For information about this choice, including consent withdrawal, please see our Privacy Policy

Reviewer #1: No

Reviewer #2: No

---

## [Author Response · Author response to Decision Letter 1]

22 Jan 2026

A detailed response to all reviewer and editor comments is provided in the attached “Response to Reviewers” document.

---

## [Decision Letter · Decision Letter 1]

5 Feb 2026

Neuromuscular Screening and Cognitive Function in Older Adults: A Cross-Sectional Exploratory Study

PONE-D-25-47478R1

Dear Dr. Adrian Agricola,

We’re pleased to inform you that your manuscript has been judged scientifically suitable for publication and will be formally accepted for publication once it meets all outstanding technical requirements.

Kind regards,

Hiroki Annaka

Academic Editor

PLOS One

Additional Editor Comments (optional):

Reviewers' comments:

Reviewer's Responses to Questions

**Comments to the Author**

Reviewer #1: All comments have been addressed

2. Is the manuscript technically sound, and do the data support the conclusions?

Reviewer #1: Yes

3. Has the statistical analysis been performed appropriately and rigorously?

Reviewer #1: N/A

4. Have the authors made all data underlying the findings in their manuscript fully available?

Reviewer #1: Yes

5. Is the manuscript presented in an intelligible fashion and written in standard English?

Reviewer #1: Yes

Reviewer #1: (No Response)

**Do you want your identity to be public for this peer review?** For information about this choice, including consent withdrawal, please see our Privacy Policy

Reviewer #1: No

---

## [Editor Report · Acceptance letter]

PONE-D-25-47478R1

PLOS One

Dear Dr. Agricola,

I'm pleased to inform you that your manuscript has been deemed suitable for publication in PLOS One. Congratulations! Your manuscript is now being handed over to our production team.

Kind regards,

on behalf of

Dr. Hiroki Annaka

Academic Editor

PLOS One